# An essential role of acetyl coenzyme A in the catalytic cycle of insect arylalkylamine N-acetyltransferase

Chu-Ya Wu [ORCID] [1,5], I-Chen Hu [ORCID] [1,5], Yi-Chen Yang[1], Wei-Cheng Ding [ORCID] [1], Chih-Hsuan Lai[1], Yi-Zong Lee[1,2], Yi-Chung Liu [ORCID] [3], Hui-Chun Cheng[1] & Ping-Chiang Lyu [ORCID] [1,4 ✉]

Acetyl coenzyme A (Ac-CoA)-dependent N-acetylation is performed by arylalkylamine N-acetyltransferase (AANAT) and is important in many biofunctions. AANAT catalyzes N-acetylation through an ordered sequential mechanism in which cofactor (Ac-CoA) binds first, with substrate binding afterward. No ternary structure containing AANAT, cofactor, and substrate was determined, meaning the details of substrate binding and product release remain unclear. Here, two ternary complexes of dopamine N-acetyltransferase (Dat) before and after N-acetylation were solved at 1.28 Å and 1.36 Å resolution, respectively. Combined with the structures of Dat in apo form and Ac-CoA bound form, we addressed each stage in the catalytic cycle. Isothermal titration calorimetry (ITC), crystallography, and nuclear magnetic resonance spectroscopy (NMR) were utilized to analyze the product release. Our data revealed that Ac-CoA regulates the conformational properties of Dat to form the catalytic site and substrate binding pocket, while the release of products is facilitated by the binding of new Ac-CoA.

[1] Institute of Bioinformatics and Structural Biology, National Tsing Hua University, Hsinchu 30013, Taiwan. [2] Instrumentation Center, National Tsing Hua University, Hsinchu 30013, Taiwan. [3] Institute of Population Sciences, National Health Research Institutes, Zhunan 35053, Taiwan. [4] Department of Medical Sciences, National Tsing Hua University, Hsinchu 30013, Taiwan. [5] These authors contributed equally: Chu-Ya Wu, I-Chen Hu. ✉email: pclyu@mx.nthu.edu.tw

Arylalkylamine N-acetyltransferase (AANAT) usually uses acetyl coenzyme A (Ac-CoA) as an acetyl donor and catalyzes the N-acetylation of arylalkylamines, such as indolethylamines and phenylethylamines[1,2] (Fig. 1). AANATs follow a sequential binding mechanism, in which cofactor (acetyl donor) binding facilitates substrate (acetyl acceptor amine) binding[3]. AANATs have been identified from a variety of vertebrates and invertebrates[4–8]. The sequence homology of AANATs is low, usually 20–40% identity from insect to insect, while there is only 13% identity between insect and vertebrate (Supplementary Fig. 1). However, the basic structures of members in the AANAT family are similar and form conserved motifs connected in the order C–D–A–B (Supplementary Fig. 1). Motifs C and D participate in maintaining the structure and stability of GNAT proteins[9]. Motif A is mainly for Ac-CoA recognition and binding[9–11], while Motif B contains some variable acidic residues for substrate binding[11].

The first solved AANAT structures were the crystal structures of serotonin N-acetyltransferase (SNAT) from sheep (Oris aries) in the apo form (PDB code: 1B6B)[12], and in a complex with a bisubstrate analog (PDB code: 1CJW)[13]. Vertebrate AANAT controls the rhythmic production of melatonin in the pineal gland[14], thereby playing a unique role in biological timing in vertebrates[2,15]. Through conversion of serotonin to N-acetyl-serotonin, the precursor of the neuro-hormone melatonin, AANAT (SNAT) can regulate the circadian rhythm and is commonly referred to as "the timezyme"[2]. The complex structure of SNAT with bisubstrate analog reveals that the conserved histidines do not directly participate in the deprotonation of the substrate amine group, but the water molecules between the conserved histidines and substrate could act as mediators for the exchangeable alkylamine proton and facilitate the nucleophilic attack of deprotonated amine on Ac-CoA[13]. In addition, Tyr168 serves to correctly position the two substrates for catalysis and offers a proton for the thiolate anion of CoA after acetyl transfer[13]. Other reports indicated that the rate-determining step in the entire reaction of N-acetylation is the diffusion of the released product[3], in which

the acetyl-substrate is released first followed by the release of CoA.

Insect AANATs play multiple physiological regulatory roles, participating not only in the inactivation of neurotransmitters[16,17] but also in cuticle sclerotization[18–20] and pigmentation[21,22]. In contrast to vertebrates, insects have multiple AANATs to regulate the metabolism of aromatic monoamines[23–25] and the biosynthesis of fatty acid amide[26–28]. Eight and 13 putative AANATs have been found in Drosophila melanogaster[25] and Aedes aegypti[29], respectively. Acetylated dopamine generated by AANAT was considered a key component in the cuticle-sclerotization process. Knockdown of AANAT resulted in malformed exoskeleton and deposition of melanin[20,30], and therefore insect AANATs were considered potential targets for insecticide design[23,24,31]. Dopamine N-acetyltransferase (Dat) is the first identified insect AANAT from Drosophila melanogaster[32,33]. The natural substrates of Dat are dopamine and tyramine, which were first proposed by Dewhurst et al.[17] using Drosophila nervous tissues and whole-fly extracts. Researchers also found that isolated Dat from Drosophila melanogaster can catalyze tryptamine, dopamine, and serotonin[33], and recombinant Dat can catalyze diverse monoamines including tyramine, dopamine, octopamine, tryptamine, and phenylethylamine (PEA) with high affinity[25]. Crystal structures of insect AANATs from Aedes aegypti (mosquito)[29,34], Drosophila melanogaster (fruit fly)[35], and Tribolium castaneum (red flour beetle)[24] have been determined. The structures of aaAANAT2, aaNAT5b, and paa-NAT7 from Aedes aegypti (PDB code: 4FD6, 5YAG, and 4FD7) are in apo forms only[29,34], and TcAANAT0 from Tribolium castaneum is in Ac-CoA-bound binary form (PDB code: 6V3T)[24], while structures of both the apo form (PDB code: 3V8I) and the Ac-CoA-bound binary form (PDB code: 3TE4) of Dat have been determined[35]. Notably, insect AANATs have an extra loop–helix–turn–helix between β3 and β4 (Supplementary Fig. 1), the "insect-specific-insert", which forms a deeper tunnel-like cavity than that in SNAT (Supplementary Fig. 2). The tunnel-like cavity in insect AANATs is composed of hydrophobic residues, and the deeply buried site is predicted to be the substrate-binding pocket[35]. In addition, the mechanism observed in insect AANATs is different

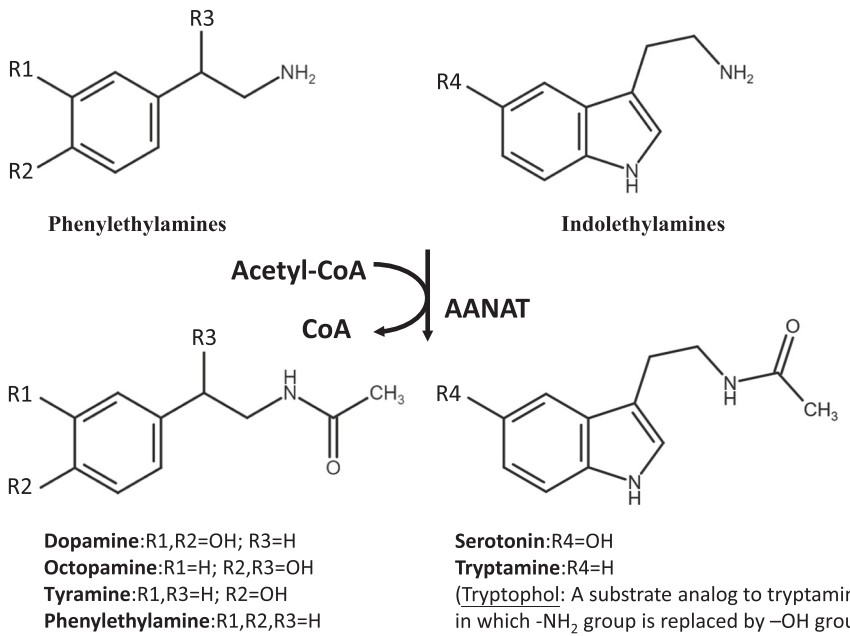

**Phenylethylamines**

**Indolethylamines**

**Acetyl-CoA**

**CoA**

**AANAT**

**Dopamine**:R1,R2=OH; R3=H
**Octopamine**:R1=H; R2,R3=OH
**Tyramine**:R1,R3=H; R2=OH
**Phenylethylamine**:R1,R2,R3=H

**Serotonin**:R4=OH
**Tryptamine**:R4=H
(Tryptophol: A substrate analog to tryptamine in which -NH₂ group is replaced by −OH group)

**Fig. 1 N-acetylation of monoamines by AANATs.** AANATs usually catalyze arylalkylamines, including phenylethylamines and indolethylamines. Dopamine, octopamine, tyramine, and phenylethylamine belong to phenylethylamines, and their structures are based upon the phenylethylamine structure as the core structure with substituents replacing one or more hydrogens. Serotonin and tryptamine belong to indolethylamines.

from that in SNAT. Different residues are in the position corresponding to the catalytic residue Tyr168 in SNAT, whereas Glu47 on α1 of Dat is proposed to be the active site residue that initiates the reaction[25,35]. Cheng et al.[35] proposed a mechanism using a catalytic triad, Glu47, Ser182, and Ser186, which are highly conserved in insect AANATs (Supplementary Fig. 1). N-acetylation includes the following steps: (i) the carboxylate anion of Glu47 serves as a general base to extract a proton from the hydroxy group of Ser182; (ii) the hydroxylate of Ser182 then acts as a general base to deprotonate the amino group of the arylalkylamine substrate; (iii) the deprotonated substrate then acts as a nucleophile to attack the carbon atom of the acetyl carbonyl of Ac-CoA to form a tetrahedral intermediate; and (iv) the hydroxy group of Ser186 could serve as a general acid to provide a proton for the thiolate anion of the CoA[35]. Dempsey et al. proposed a simpler pathway initiated with Glu47 capturing the proton of the closest water molecule and facilitating the deprotonation of substrate[25].

Although it is well established that the acetyl transfer reaction performed by AANATs occurs through an ordered sequential mechanism, the details of substrate binding and product release remain unclear. Among the known structures of AANATs, only SNAT has been solved in complex with a bisubstrate analog, and no ternary complex with cofactor and substrate is known. This means that the exact location of the substrate-binding site in insect AANATs is unknown. The cavity predicted to be the substrate-binding pocket in insect AANATs is longer than that in SNAT, indicating that the mechanism of substrate binding and product release in insect AANATs may be different from that in SNAT. In addition, the role of the acetyl group in driving the formation of ternary complexes before and after catalysis remains unknown. In this work, we report the first crystal structures of the ternary complex of insect AANAT, to our knowledge. Two ternary forms, Dat/Ac-CoA/tryptophol (analog of tryptamine) and Dat/CoA/acetyl-tryptamine (product of tryptamine, Ac-TRYP), were determined in order to elucidate the structural difference before and after acetyl transfer. The crystal structure of the Dat/CoA complex was also determined to help identify the role of the acetyl group in the catalytic cycle. A combination of isothermal titration calorimetry (ITC) and nuclear magnetic resonance spectroscopy (NMR) was used to investigate the structure–activity relationship during the catalytic cycle of Dat. Our data reveal that Ac-CoA plays a critical role in regulating the conformational properties to form the catalytic site and substrate-binding pocket as well as in promoting product release and initiating the next round of catalysis. These structural details presented here broaden our understanding of the structural features required for insect AANATs to perform the catalytic cycle and provide an alternative model for investigations of other AANATs.

## Results

**Structural basis for Dat substrate and product binding.** *Overall structures of ternary complexes*: The first ternary complexes of insect AANATs are presented here. To compare the structures before and after catalysis, two ternary forms, Dat/Ac-CoA/tryptophol (mimicking the enzyme-substrates complex) and Dat/CoA/Ac-TRYP (representing the enzyme-products complex), were obtained through cocrystallization and solved at resolutions of 1.28 and 1.36 Å, respectively. Data collection and refinement statistics are listed in Table 1. Tryptophol is a dead-end analog of tryptamine for N-acetyltransferase because the amino group in tryptamine is replaced as a hydroxyl group to form tryptophol. The two ternary structures before and after catalysis were very similar (RMSD = 0.65 for all atoms) (Fig. 2a). Overall structures revealed that Dat was a globular protein containing a tunnel-like

cavity, a substrate located in the middle of the protein, a CoA group located at the bottom, and an acetyl group between the substrate and CoA (Fig. 2a). The locations of indole rings in tryptophol and Ac-TRYP were almost the same, while the locations of two acetyl groups in Ac-CoA and Ac-TRYP were very similar.

*Substrate-binding pocket*: The substrate-binding pocket was a hydrophobic pocket composed of the α1 terminus, α2, α4, ß4, and ß5 (Fig. 2b). The backbone carbonyl oxygen of Leu180 may form a hydrogen bond with the active nitrogen atom in Ac-TRYP (Fig. 2b; Supplementary Table 1), while a hydrogen bond with the hydroxyl group of tryptophol was also possible. Therefore, we considered that the role of this hydrogen bond was to position the active nitrogen atom in the substrate. The backbone amide hydrogen of Leu146 also formed a hydrogen bond with acetyl groups either in product (Ac-TRYP) or cofactor (Ac-CoA) (Fig. 2b; Supplementary Table 1). We also proposed that the role of this hydrogen bond was to position the acetyl group. Three aromatic residues, Phe43, Tyr64, and Phe114, were around the indole ring of Ac-TRYP (Fig. 2b; Supplementary Table 1). According to the distance and orientation of these aromatic rings, π–π interactions may occur among them to stabilize the bound substrate. An interaction between the sulfur atom of methionine (Met) and the aromatic rings of phenylalanine (Phe), tryptophan (Trp), or tyrosine (Tyr) was recently proposed[36]. Met-aromatic interaction was identified as a critical factor in the stabilization of the protein structure because this interaction yielded an additional stabilization of 1–1.5 kcal/mol compared with a purely hydrophobic interaction[36]. In the ternary complexes, Met-aromatic interaction was proposed between Met121 and Tyr64 and between Met121 and Phe114 (Fig. 2b). This interaction may be important to stabilize the substrate-binding pocket.

*Cofactor-binding pocket and the catalytic site*: The cofactor (Ac-CoA or CoA)-binding pocket was formed by α1, α5, α6, α7, ß4, and ß5 (Fig. 2c). The CoA group directly formed hydrogen bonds with the backbone atoms on ß4, α5, and α6 and built two salt bridges with Lys192 (Fig. 2c; Supplementary Table 1). Notably, backbone atoms of Leu146 can interact with both the acetyl group and CoA group (Fig. 2c); therefore, the backbone of this residue was important in positioning Ac-CoA. The locations of previously reported catalytic residues, Glu47 (on α1), Ser182 (on the loop before α7), and Ser186 (on α7)[35], were not altered before and after catalysis. (Fig. 2c).

**Conformational differences in each stage of catalysis.** *Overall structures*: The structural alignments of the apo form (initial stage), binary complex (cofactor-binding stage), ternary complex with Ac-CoA and substrate analog (substrate-binding stage), and ternary complex with products (product-binding stage) of Dat revealed that the whole structures were not much different, but some local conformations of apo-Dat were recognizably different (shown by arrows in Fig. 3a). The central ß-sheets (ß3–5 and ß7) were almost fixed, while the peripheral α-helices, such as α1 terminus, α2, and α5–7, had moved significantly. Between the initial stage and the cofactor-binding stage, the following significant differences were present: the considerable shift of a region from Leu146 to Ala158 (including C-terminus of β4, α5 and N-terminus of α6) toward the bound cofactor, the appearance of a new helix α9 consisting of five residues, Ala217, Ala218, Pro219, His220, and Val221, the large movement of α7, and the shift of α1 (Fig. 3a). An all-atom-based RMSD from the conformationally changed regions between apo-Dat and the cofactor-binding stage was 2.24 Å, while those between the cofactor-binding stage and two ternary complexes before and after N-acetylation were 0.69 Å and 0.60 Å, respectively.

**Table 1 Data collection and refinement statistics of the complex structures of Dat.**

| Structure | Dat/Ac-CoA/tryptophol | Dat/CoA/Ac-TRYP | Dat/CoA |
|---|---|---|---|
| PDB code | 6K80 | 5GI9 | 5GI5 |
| Data collection | | | |
| Space group | $P2_12_12_1$ | $P2_12_12_1$ | $P2_12_12_1$ |
| Cell dimensions | | | |
| a, b, c (Å) | 43.64, 56.45, 84.07 | 43.50, 56.46, 84.36 | 47.84, 47.78, 90.69 |
| α, β, γ (°) | 90, 90, 90 | 90, 90, 90 | 90, 90, 90 |
| Resolution (Å) | 30–1.28 (1.33–1.28) | 30–1.36 (1.41–1.36) | 30–1.42 (1.47–1.42) |
| $R_{p.i.m.}$ (%) | 1.9 (23.0) | 2.4 (28.6) | 1.6 (21.3) |
| I/σ | 39.2 (2.7) | 31.6 (1.7) | 41.4 (2.1) |
| Completeness (%) | 99.5 (95.8) | 99.8 (99.5) | 99.6 (98.1) |
| Redundancy | 7.5 (5.3) | 6.7 (5.7) | 10.3 (9.4) |
| Refinement | | | |
| Resolution (Å) | 26.76–1.28 (1.33–1.28) | 28.23–1.4 (1.45–1.4) | 25.55–1.45 (1.50–1.45) |
| Number of reflections | 54,013 (5056) | 41,602 (4097) | 37,435 (3617) |
| $R_{work}/R_{free}$ | 18.5/19.9 (24.0/26.8) | 15.5/18.5 (17.7/24.0) | 15.5/18.7 (16.4/21.7) |
| Number of atoms | | | |
| Protein | 1712 | 1688 | 1672 |
| Ligand | 87 | 86 | 53 |
| Water | 261 | 199 | 197 |
| B-factor | | | |
| Protein | 15.7 | 20.6 | 22.3 |
| Ligand | 14.3 | 18.4 | 15.8 |
| Water | 23.8 | 31.2 | 31.2 |
| RMSD | | | |
| Bond length (Å) | 0.006 | 0.007 | 0.006 |
| Bond angle (°) | 1.24 | 1.20 | 1.17 |
| Ramachandran plot | | | |
| Favored regions (%) | 97.6 | 90.7 | 91.6 |
| Allowed regions (%) | 2.4 | 8.7 | 8.4 |
| Outliers | 0.00 | 0.5 | 0 |

*The closed form of the cofactor-binding pocket*: In the cofactor-binding stage, considerable shifts of the C-terminus of β4, α5 and the N-terminus of α6 (including the connecting loops) toward the bound cofactor were observed (Fig. 3a). This moved region (Leu146–Ala158) extends into the groove between the 3′-adenosine phosphate and diphosphate parts of Ac-CoA (Fig. 3b) and interacted with them through several hydrogen bonds (Supplementary Table 2). An 84.1° rotation of the side chain of Asp46 on the shifted α1 brought its carboxylate group toward the guanidinium group of Arg153 on the shifted α5 (which shifted 1.9 Å toward α1) (Fig. 3b). The distance between the side chains of Asp46 and Arg153 was shortened from 5.0 Å in the apo form to 2.7 Å in the cofactor-bound forms (Supplementary Table 3), and then it was within the distance for salt-bridge interaction. In addition, the side chain of Lys192 on the end of α7 was rotated by 111.2°, creating two salt bridges to the diphosphate group and the 3′-phosphate group of the CoA moiety to stabilize the bound cofactor (Fig. 3b). The region (Leu146–Ala158) served as a hook, moving from an open position to a closed position, to keep the bound cofactor inside the cofactor-binding pocket. Therefore, we also descripted that Ac-CoA binding caused the cofactor-binding pocket to be converted into a closed form.

*Catalytic site*: The newly formed α9 was located near α7, in which two catalytic residues, Ser182 and Ser186, resided (Fig. 3c). Two interhelix hydrogen bonds were formed between the newly formed α9 and the moved α7 (side chains between His220 and Ser183, as well as backbones between His220 and His184), while a stacking interaction between His220 on α9 and Tyr185 on α7 was observed (Fig. 3c and the distances shown in Supplementary Table 3). Both the main chain and side chain of Ser182 and Ser186 shifted as α7 moved (Fig. 3c). The shift of α1 changed the orientation of the other catalytic residue Glu47 (rotated by 10.6°)

and moved the side chain of Glu47 closer to the side chain of Ser182 (Fig. 3c). The distance between Glu47 and Ser182 decreased from 3.3 Å in the initial stage to 2.6 Å in the cofactor-binding stage (Supplementary Table 3). The above-mentioned conformational changes indicated that the catalytic site was built in the cofactor-binding stage.

*Substrate entrance and binding pocket*: Dat contained a tunnel-shaped cavity, and the cofactor occupied one side of this cavity (Fig. 2a). Considering the ordered sequential mechanism of Dat, we supposed that the substrate could use the entrance on the other side to pass through the tunnel and arrive at the catalytic site in the middle of Dat. Ac-CoA binding resulted in significant changes in the shape of the cavities (Fig. 3d). A bottleneck (the narrowest site) in the cavity surrounded by Met121 and Asp142 was also created after Ac-CoA binding (radius of 1.9 Å) and changed to become wider after substrate binding (radius of 2.4 Å)[37]. The size change of the bottleneck of the cavity might be related to the selectivity of substrate binding and the release of products. Based on the structures of ternary complexes, Met121 and Tyr64 as well as Met121 and Phe114 may induce Met-aromatic interactions to stabilize the substrate-binding pocket (Fig. 2b). In addition, Tyr64, Phe114, and Phe43 were considered to stabilize the bound substrate by π–π interactions with their aromatic rings (Fig. 2b). The locations of these three residues in the cofactor-bound forms (binary and ternary complexes) were almost the same, but recognizably different in the initial stage (Figs. 2b and 3e). Slight movements of α1 and α4 resulted in shorter distances between the rings of the substrate and Phe43 and Phe114 in the cofactor-bound forms. However, a shift of α2 in the cofactor-bound forms along its helical axis by approximately one-quarter of a helical turn toward the C-terminus increased the distance from Tyr64 to the indole ring of Ac-TRYP. The distances of the sulfur atom in Met121 to aromatic rings in Tyr64 and Phe114 were 7.5 and 9.4 Å in

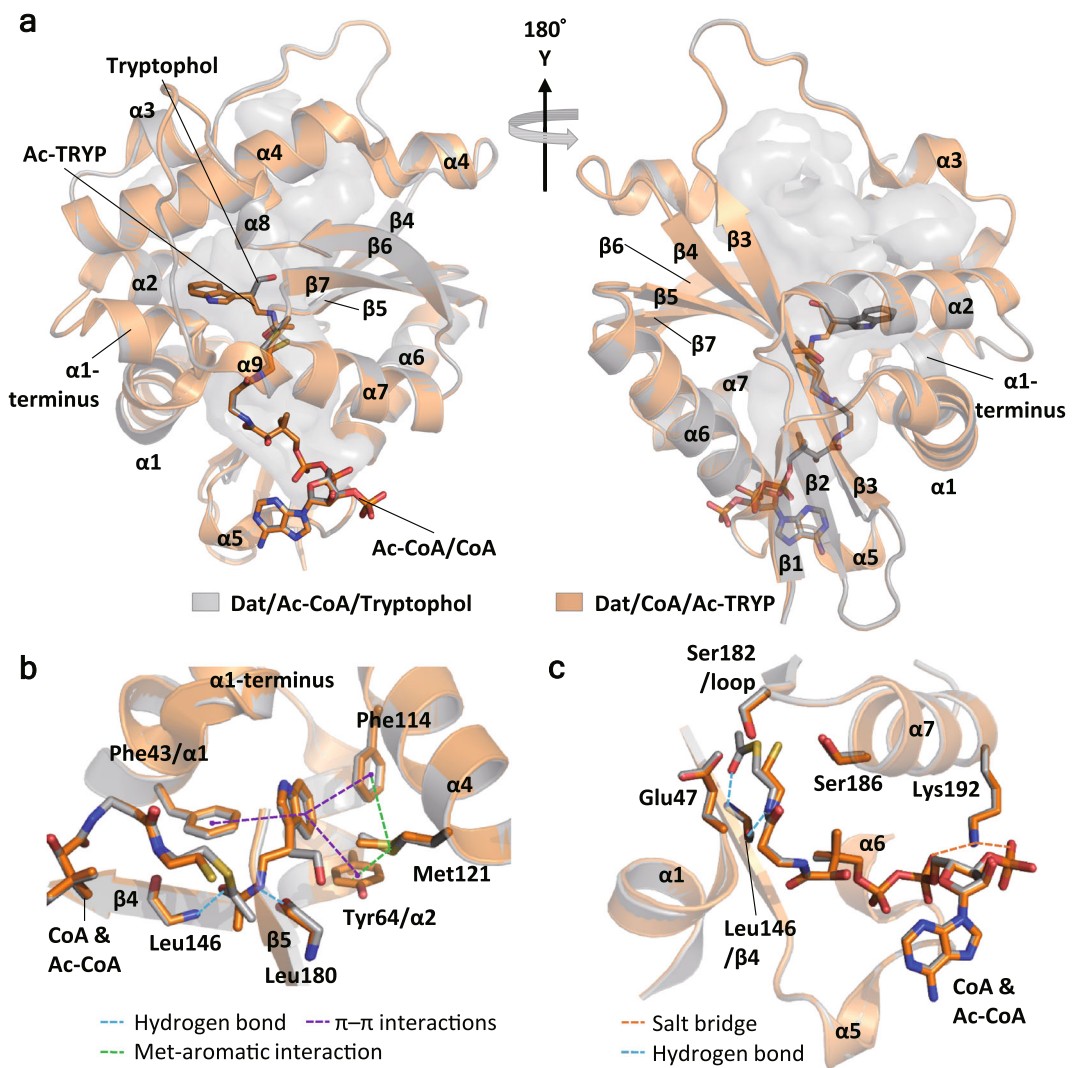

**Fig. 2 The enzyme–substrate-mimicking complex (Dat/Ac-CoA/tryptophol) and enzyme-product complex (Dat/CoA/Ac-TRYP) have similar structures. a** Superposition of Dat/Ac-CoA/tryptophol (gray, substrate-mimicking complex, PDB code: 6K80) and Dat/CoA/Ac-TRYP (orange, product complex, PDB code: 5GI9). The Cα-based RMSD of the two structures is 0.10 Å (183 atoms), and the all-atom-based RMSD is 0.65 Å (1688 atoms). Ac-CoA, tryptophol, CoA, and Ac-TRYP are shown as sticks. The solvent-accessible tunnel is shown in gray. **b** The aromatic residues of the substrate-binding pocket. The distances of Met121 to Tyr64 and Phe114 (green dashed line) were 5.5 and 7.0 Å, respectively. The distances from the center of the indole ring of the substrate to Phe43, Tyr64, and Phe114 (purple dashed line) were 5.0, 5.8, and 5.8 Å, respectively. **c** Cofactor-binding site and catalytic triad.

apo-Dat, as well as 5.3 and 7.0 Å in the Dat/Ac-CoA complex (Fig. 3e; Supplementary Table 3). According to these distances, Met-aromatic ring interactions occurred in the cofactor-bound forms but not in apo-Dat. Because there were no significant changes among the cofactor-binding stage, substrate-binding stage, and product-binding stage, but they were different from initial stage, we proposed that Ac-CoA binding triggers the conformational change to create the substrate-binding pocket.

**The distinctive roles of the acetyl group and the CoA moiety.** The above structural comparison revealed that cofactor binding is important for the initiation of the catalytic cycle because the substrate-binding pocket and the catalytic site were created in the cofactor-binding stage. To distinguish the roles of the acetyl group and the CoA moiety in the regulation of conformational changes, the crystal structure of the Dat/CoA complex was solved at a resolution of 1.45 Å as a control to compare with the Dat/Ac-CoA complex. The data collection and refinement statistics for the Dat/CoA complex are listed in Table 1. In the Dat/CoA

complex, there was no additional α9, and the position of α7 was different (Fig. 4), indicating that the formation of α9 and the shift of α7 occurred in the presence of the acetyl group. Because the interactions between α9 and the shifted α7 were critical for catalytic site (Fig. 3c), this result also supported that the acetyl group fine-tuned the catalytic site through forming α9 to interact with α7. In contrast, the locations of α5 and the critical salt-bridge interactions of Asp46–Arg153 and Lys192–CoA moiety were very similar in these two binary complexes (Fig. 4), indicating that the conformational changes for stabilizing bound cofactor were regulated by the CoA moiety.

**Ac-CoA binding is required for releasing products.** In an attempt to obtain intermediate structures during acetyl transfer, we soaked Dat/Ac-CoA crystals in substrate solution before freezing the crystals. Both TRYP and PEA were tested, but only products, CoA and acetyl substrate, were observed in the final crystal structures. Cocrystallization of Dat, Ac-CoA, and PEA also yielded crystals of Dat with two products, CoA and Ac-PEA (PDB

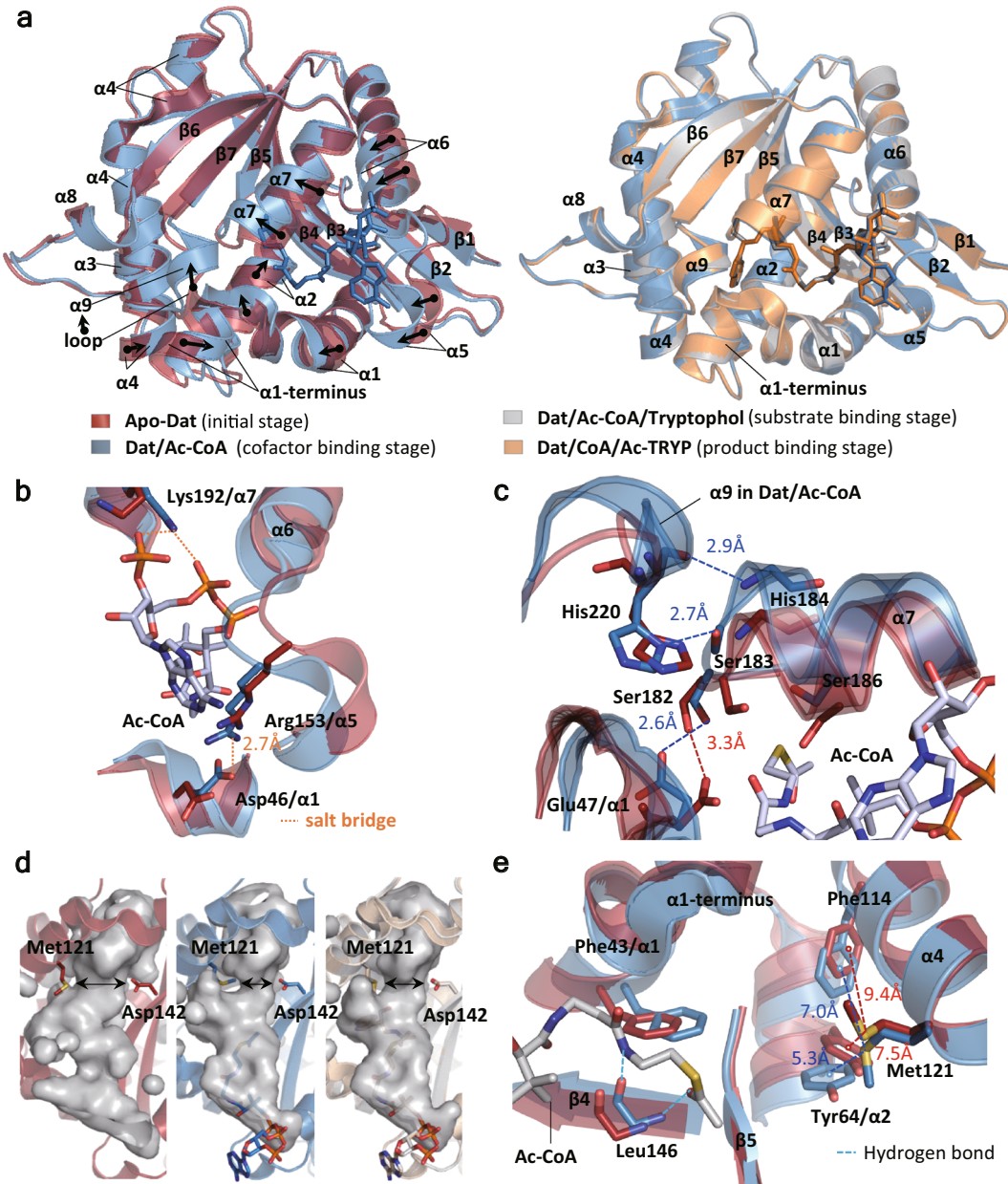

**Fig. 3 Conformational comparison in different stages of catalysis. a** Overall structural comparison of Dat. Left, superimposition of apo-Dat (red, PDB code: 3V8I) and Dat/Ac-CoA complex (blue, PDB code: 3TE4); right, superimposition of Dat/Ac-CoA complex (blue, PDB code: 3TE4), Dat/Ac-CoA/ tryptophol complex (gray, PDB code: 6K80), and Dat/CoA/Ac-TRYP complex (orange, PDB code: 5GI9). Conformational changes are shown by arrows. **b** Cofactor-binding pocket showed an open-to-closed form conversion from apo-Dat (red) to Dat/Ac-CoA complex (blue), forming salt bridges in α1/α5/ α7 region. **c** Superimposition of apo-Dat (red) and Dat/Ac-CoA complex (blue) focusing on catalytic site. Two hydrogen bonds between α7 and α9 were established by His220, Ser183, and His184 after Ac-CoA binding. The orientations of catalytic residues, Glu47, Ser182, and Ser186 also changed. **d** The reaction tunnels in the apo-Dat, binary complex, and two ternary complexes are shown in gray. A bottleneck surrounded by Met121 and Asp142 was formed after Ac-CoA binding. **e** Superimposition of apo-Dat (red) and Dat/Ac-CoA complex (blue) focusing on aromatic residues of the substrate-binding pocket. Residues, substrates, and cofactors are shown as sticks. Distances, salt-bridge interactions, hydrogen bonds, and Met-aromatic interactions are shown by dashed lines.

code: 5GI7). These results indicated that the acetyl transfer was rapid, and the products could not release from Dat. These findings also prompted us to investigate how the next catalytic cycle begins. Considering the ordered binding mechanism of Dat, we hypothesized that the addition of Ac-CoA to the Dat-products complex would competitively replace the bound products and restart the next round of the catalytic cycle. The replacement of the bound CoA by Ac-CoA is possible because the $K_d$ of Ac-CoA binding to Dat ($5.51 \pm 1.92\,\mu M$) is ~30-fold lower than that of

CoA ($152.89 \pm 26.48\,\mu M$) (Fig. 5 and Table 2). To test our hypothesis, the titration of Ac-CoA into the product-bound complex, Dat/CoA/Ac-PEA, was performed, and a significant heat change was detected by ITC (Fig. 5c). A competition assay using PEA substrate against the bound products in Dat was also performed, but no significant heat change was observed (Fig. 5d). These competition assay results indicated that Ac-CoA could competitively replace the bound CoA in Dat, but PEA did not.

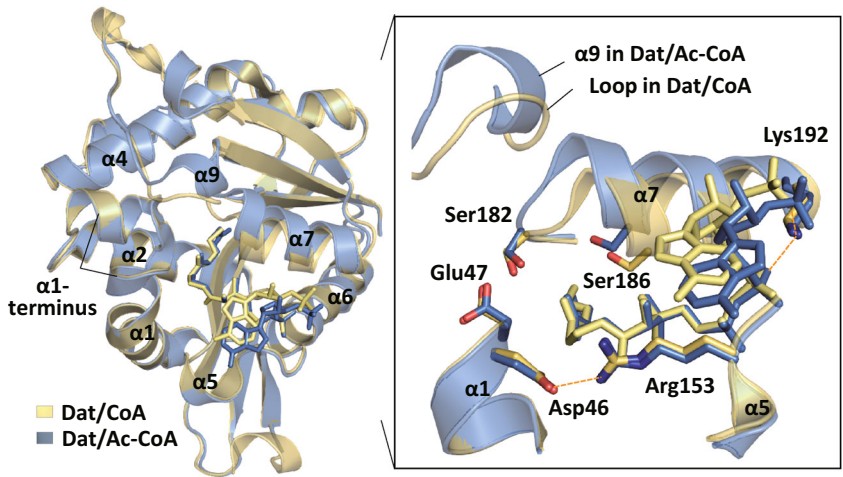

**Fig. 4 Active site is fine-tuned by the acetyl group.** Structural comparison of Dat/CoA complex (yellow, PDB code: 5GI5) and Dat/Ac-CoA complex (blue, PDB code:3TE4) reveals different positions of α7 and loop–α9 changes. Residues are shown as sticks. Salt bridges in Asp46–Arg153 and Lys192–CoA moiety are shown in orange dashed lines.

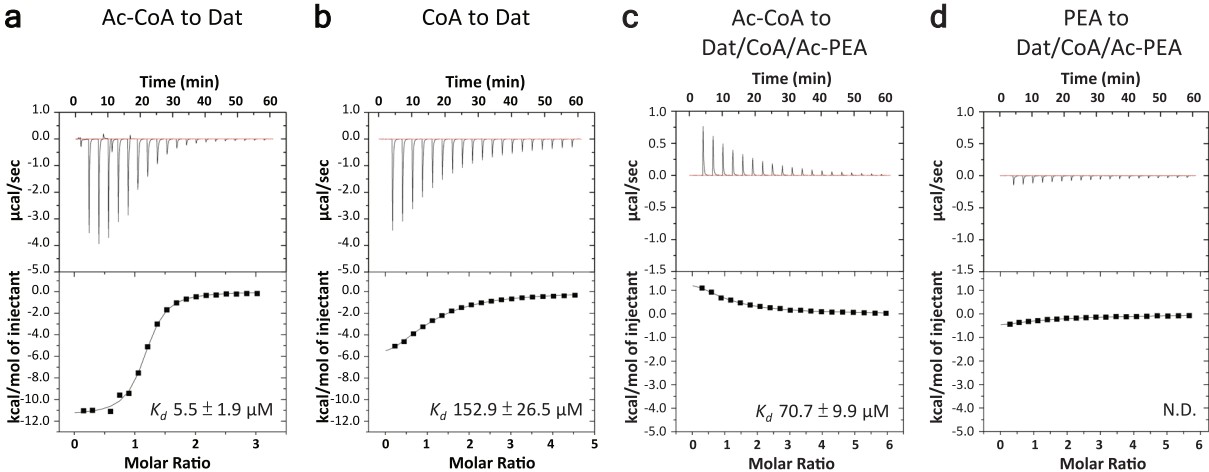

**Fig. 5 Investigation of Dat catalytic cycle by ITC experiment. a, b** ITC-binding study on Ac-CoA to apo-Dat (**a**), and CoA to apo-Dat (**b**). **c, d** ITC-titration experiment of Ac-CoA against the Dat/CoA/Ac-PEA ternary complex (**c**), and PEA against the Dat/CoA/Ac-PEA ternary complex (**d**). $K_d$ values were calculated with $n = 6$ biologically independent experiments. N.D. data cannot be determined.

| Table 2 ITC analysis of acetyl-CoA, CoA, PEA, and Ac-PEA binding against Dat. | | | | | | | |
|---|---|---|---|---|---|---|---|
| Group[*] | In syringe | In sample cell | $N$ | $Kd$ (µM) | $\Delta H$ (kcal/mol) | $-T\Delta S$[**] (kcal/mol) | $\Delta G$[***] (kcal/mol) |
| | Ligand | Protein | | | | | |
| a | Ac-CoA | Dat | 1.11 | 5.51 ± 1.92 | −12.33 ± 2.37 | 5.09 ± 2.29 | −7.24 ± 0.25 |
| b | CoA | Dat | 1.11 | 152.89 ± 26.48 | −9.35 ± 1.43 | 4.12 ± 1.46 | −5.23 ± 0.10 |
| c | Ac-CoA | Dat/CoA/Ac-PEA | 1.86 | 70.7 ± 9.90 | 1.38 ± 0.11 | −7.06 ± 0.16 | −5.67 ± 0.08 |
| d | PEA | Dat/CoA/Ac-PEA | cannot be determined | | | | |

[*]Groups a–d corresponded to Fig. 5a–d.
[**]$T\Delta S$ – entropy variation; calculated from the equation $\Delta G = \Delta H - T\Delta S$.
[***]$\Delta G$ values were calculated using the equation $\Delta G = -RT\ln Ka$.
All values correspond to the mean of experiments repeated with $n = 6$ biologically independent experiments.

**Visualizing the release of products in crystallo and in solution.** The ejection of the two products by Ac-CoA was further examined by X-ray crystallography. We performed the competition reaction by soaking the Dat/CoA/Ac-PEA crystal in 20 mM Ac-CoA for 30 min, and the diffraction data were collected immediately. Electron density for Ac-PEA disappeared, and new Ac-CoA was observed (Fig. 6a, b), but the electron density for Ac-PEA did not change in the control experiment, in which the

Dat/CoA/Ac-PEA crystal was mixed with pure water for 30 min (Fig. 6c).

[1]H-[15]N heteronuclear single quantum coherence (HSQC) spectra were further employed to monitor the behavior of Dat during the titration of Ac-CoA into the products-bound complex. The addition of Ac-CoA to [15]N-labeled Dat/CoA/Ac-PEA resulted in significant chemical shift perturbation (CSP) of a large number of resonances (Fig. 6d). When the molar ratio was

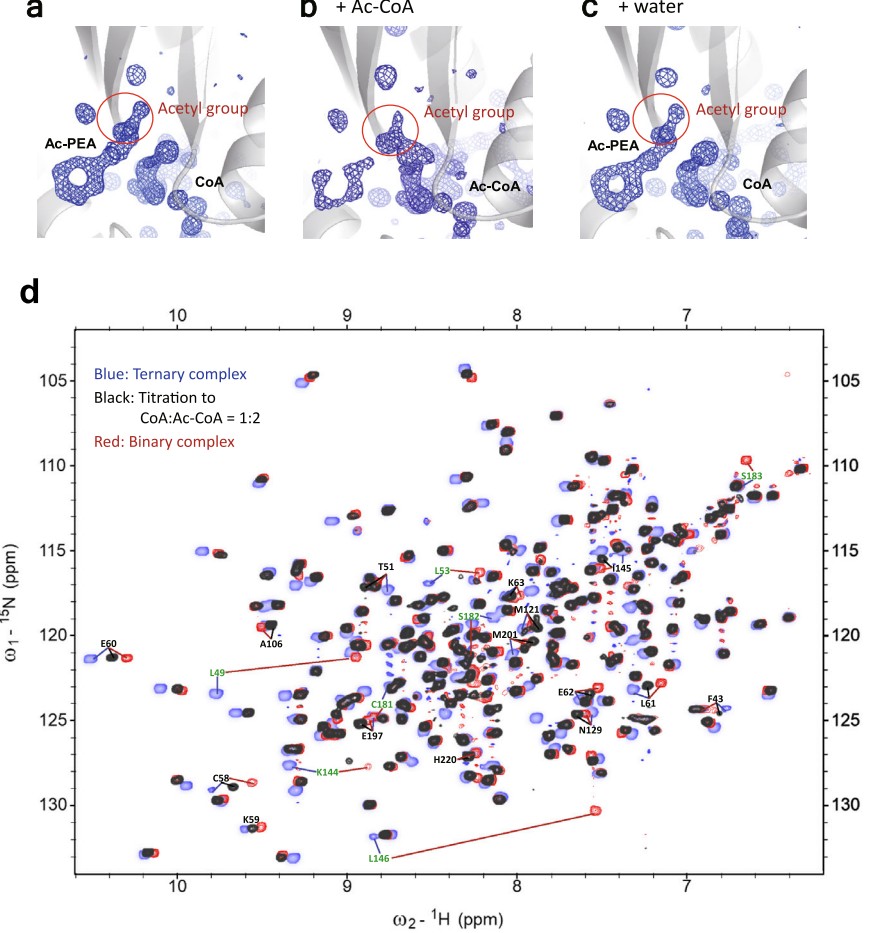

**Fig. 6 Acetyl-CoA can eliminate the products of Dat and form a new binary complex. a–c** Fo–Fc omit electron density maps contoured at 3σ for ligands in the crystal of the Dat/CoA/Ac-PEA complex (**a**), the crystal after mixing the Dat/CoA/Ac-PEA crystal with Ac-CoA solution (final concentration of 20 mM) for 30 min (**b**), and the crystal after mixing the Dat/CoA/Ac-PEA crystal with pure water for 30 min (**c**). **d** Superimposition of HSQC spectra of the Dat/CoA/Ac-PEA complex (blue), end of Ac-CoA titration (CoA:Ac-CoA = 1:2, black), and Dat/Ac-CoA complex (red). Residues that lost the resonances corresponding to the end of titration are labeled in green. Some residues that showed significant CSP between the end of titration and the binary complex are labeled in black.

higher than 1:2 (CoA:Ac-CoA), the HSQC spectrum exhibited no further change, and was highly similar to that of the Dat/Ac-CoA complex (red spectrum in Fig. 6d). Two major differences were observed. One was the loss of resonances corresponding to the end of titration (residues labeled green in Fig. 6d), while the other was significant CSP between the end of titration and the binary complex (residues labeled black in Fig. 6d). Leu146 participates in both the substrate-binding and cofactor-binding sites; its backbone amide group can form hydrogen bonds to the oxygen of the acetyl group (whether it was in the cofactor or substrate), while its backbone carbonyl oxygen atom can form hydrogen bonds with the amide hydrogen atom of the β-mercaptoethylamine moiety of Ac-CoA and CoA. The resonance of Leu146 disappeared when Ac-CoA was added to the ternary complex, and no resonance corresponding to Leu146 was found in the titration spectra. In addition, the resonance of Leu146 in the binary complex was very different from that in the ternary complex. This information could indicate that multiple states for the amide group of Leu146 existed during the titration of Ac-CoA. The results of both X-ray crystallography and NMR confirmed that newly added Ac-CoA competes with and releases the two products because of the higher affinity of Ac-CoA than that of CoA and because the acetyl group of Ac-CoA occupied the same acetyl-binding site as Ac-PEA.

## Discussion

According to the above data, we propose a complete catalytic cycle of Dat (Fig. 7). The first round of N-acetylation was triggered by Ac-CoA. This Ac-CoA binding resulted in conformational changes to stabilize the bound cofactor, fine-tune the catalytic site, and create a substrate-binding pocket. After the acetyl group transfer, two products, CoA and acetyl substrate, occupied the cofactor and the substrate-binding pockets, respectively. Adding a new Ac-CoA would replace the CoA bound with Dat and eliminate the acetylated product, after which a new binary complex would be ready for the next catalytic cycle.

*The structural switch couples the cofactor binding to the formation of substrate-binding pocket*: Our X-ray crystallographic studies supported that Ac-CoA binding triggers the conformational change and creates the substrate-binding pocket. A significant movement of the region from Leu146 to Ala158 (the C-terminus of β4 to α5 to the N-terminus of α6) toward the bound cofactor was observed after cofactor binding (Fig. 3a). This moved region (Leu146–Ala158) served as a hook to interact and stabilize Ac-CoA, while a salt bridge between Arg153 and Asp46 (on α1) formed to fix this movement (Fig. 3b). A corresponding salt bridge was also observed in SNAT (E54 and R131) in its bisubstrate complex, but not in its free state[13]. In addition, these

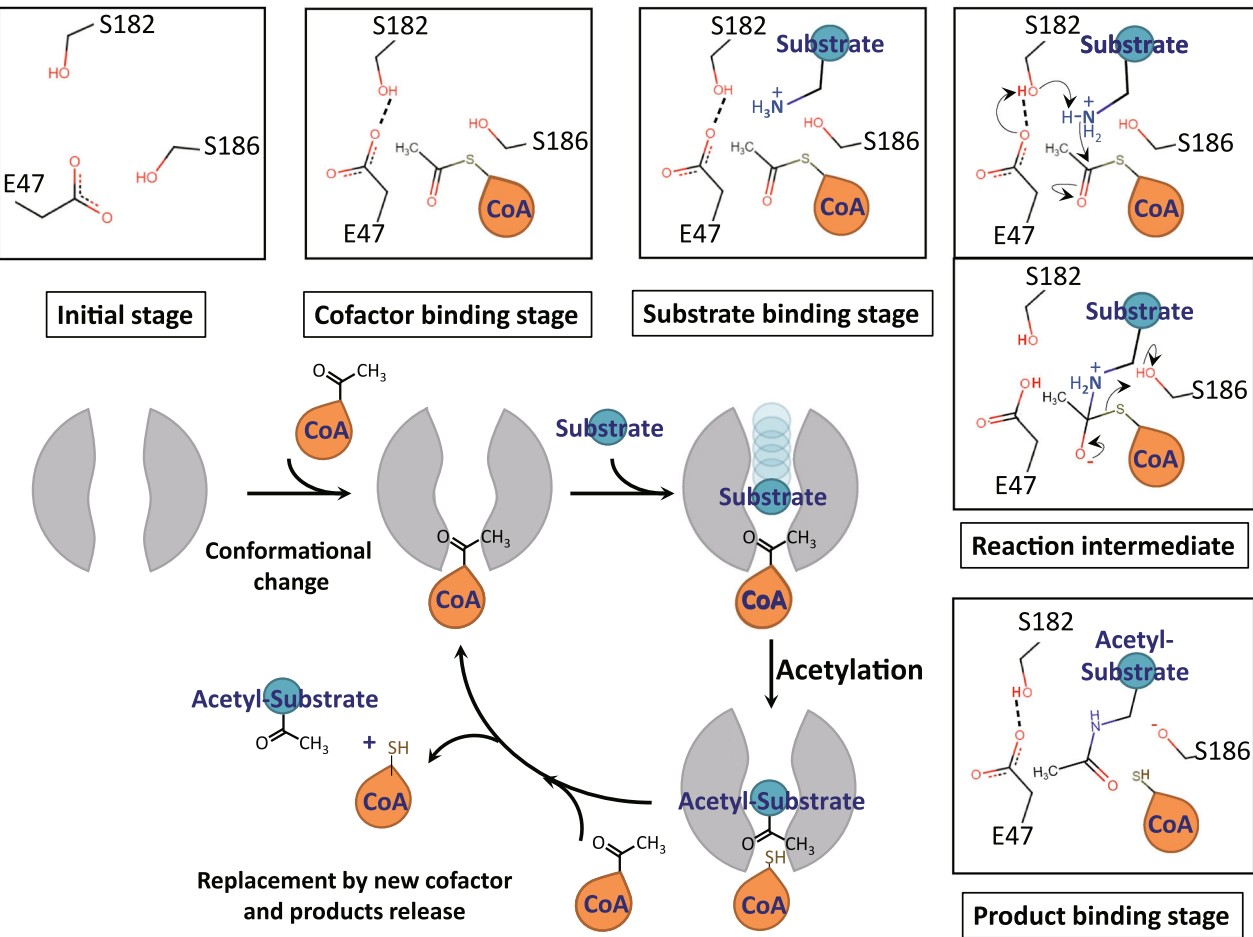

**Fig. 7 Schematic representation of the catalytic cycle of Dat.** Dat exhibits an ordered sequential mechanism in which the cofactor (Ac-CoA) binds first, followed by the binding and acetyl transfer of the substrate. Ac-CoA binding results in a conformational change to form a binding pocket for substrates. After acetyl transfer, two products (CoA and Ac-substrate) still bind in Dat and could be released through the binding of a new Ac-CoA. The new Ac-CoA could compete with the CoA bound with Dat and could release the Ac-substrate, and a new Dat/Ac-CoA complex is ready for the next catalytic cycle.

two residues were conserved in insect AANATs (Supplementary Fig. 1), suggesting their functional importance. We assumed that disruption of this structural stabilizing interaction related to Ac-CoA binding would affect the formation of the substrate-binding pocket and result in low catalytic activity. The results from two sets of the data based on mutants supported our hypothesis. Dempsey et al. reported that Ac-CoA-bound Dat R153A had a 20-fold higher $Km,app$ for tyramine[25], while Aboalroub et al. reported a 10-fold decrease in affinity for tyrosol binding to Ac-CoA-bound bmAANAT3 D26A (Asp26 of bmAANAT3 corresponds to Asp46 of Dat)[38]. Therefore, we propose that the Asp46–Arg153 salt bridge may serve as a structural switch to mediate the formation of the substrate-binding site through sensing the occupancy at the cofactor site. Because the Asp46–Arg153 salt bridge was also observed in the Dat/CoA complex (Fig. 4), we further concluded that the structural switch sensed the occupancy of the CoA moiety at the cofactor-binding site, which is different from the suggestion of Aboalroub et al. (sensing the acetyl group)[38].

*The fine adjustments of the catalytic site*: A detailed description of the catalytic site formation is proposed as follows. After Ac-CoA binding, α1 and α7, on which the catalytic residues (Glu47, Ser182, and Ser186) are located, moved. The side chain of Glu47 on α1 rotates and results in a shortened distance between Glu47 and Ser182 (on α7), while the side chain of Ser186 also rotates for

the further duty of protonating the sulfide atom of CoA. We have also identified that the acetyl group plays a key role in fine-tuning the catalytic site through forming α9 to interact with α7. The backbone amide hydrogen of Leu146 formed a hydrogen bond with acetyl groups either in cofactor (Ac-CoA) or product (Ac-TRYP) (Figs. 3e and 2b). The position of Leu146 moved significantly between the apo form and Ac-CoA bound form, but remained in the same position in the ternary complexes (Fig. 3e). Also considering that the hydrogen bond was through the backbone amide hydrogen of Leu146, therefore, we propose that residue146 is responsible for positioning the acetyl group at the catalytic site by forming a hydrogen bond between them.

*The role of the tunnel-shaped cavity*: Dat contained a tunnel-shaped cavity, and a bottleneck of the cavity surrounded by Met121 and Asp142 was created in the cofactor-binding stage but became wider after substrate binding (Fig. 3d). The size change of the bottleneck of the cavity might be related to the selectivity of substrate binding and the release of products. Since the position of Asp142 has not changed from the initial stage to the product-binding state, we supposed that Met121 may be the controller of the bottleneck. However, the alanine substituted mutant of Met121 lost part of its secondary structure (from Circular dichroism (CD) spectra) and tertiary structure (from NMR HSQC spectra) and caused an unstable structure, which was easy to aggregate (Supplementary Fig. 3). This may be due to the loss

of Met-aromatic interactions, which contribute to the stability of protein structure[36]. Therefore, we are currently unable to identify the role of Met121 by studying its mutants.

*The release of products*: Comparison of three known AANAT structures (Supplementary Figs. 1 and 2) revealed that the overall structure of apo-aaAANAT2 (from mosquitoes, PDB code: 4DF6) was similar to the ternary complex of Dat (from fruit fly, PDB code: 5GI9) but without α9, whereas significant differences were observed between SNAT (from sheep) and insect AANATs. The insect-specific-insert, the plug in the α1 terminus, α8, and α9 (or the loop in apo-aaAANAT2) made the cavities of insect AANATs much larger than that of SNAT (Supplementary Fig. 2). Hickman et al. suggested that since the product of SNAT, N-acetyl-serotonin, was more hydrophobic than serotonin, this property facilitated product departure from the binding site[13]. They further proposed that the protein in the ternary conformation with an empty substrate-binding site was unstable and could convert into an uncomplexed molecule accompanied by the ejection of CoA[13]. However, in the case of Dat, differential scanning calorimetry (DSC) data (Supplementary Fig. 4) revealed that the thermal stability of Dat/CoA was higher than that of apo-Dat and lower than that of Dat/CoA/Ac-PEA, while Tm of Dat/Ac-CoA and Dat/CoA/Ac-PEA were almost the same. In addition, the high hydrophobicity of the tunnel-shaped cavity in Dat could be more suitable for the acetyl substrate than the cavity in SNAT. Therefore, we proposed that products in Dat would be ejected by the new Ac-CoA and confirmed this proposal by ITC titration (Fig. 5), crystallography (Fig. 6b), and NMR titration (Fig. 6d). We concluded that after the first acetyl transfer, to initiate the next reaction, a new Ac-CoA is necessary to replace the CoA bound to Dat and release the acetyl substrate. At that point, a new Dat/Ac-CoA complex is ready to bind a new substrate and perform acetyl transfer again.

*The catalytic mechanism*: The final question we want to answer is the catalytic mechanism of Dat. Cheng et al. proposed a mechanism using catalytic triad, which was started by deprotonating Ser182 by Glu47 and then Ser182 deprotonating the amine group of substrate[35], while Dempsey et al. proposed a simpler pathway initiated by Glu47 capturing the proton of the closest water molecule[25]. We confirmed the water molecules in the structure at different stages of the catalytic cycle and found that the number of water molecules at the catalytic site decreased as the catalytic cycle progressed (Supplementary Fig. 5). According to the relative positions and distances (Fig. 3c and Supplementary Table 3), Ser182 should be the best candidate to serve as a mediator to transfer proton of the substrate. Therefore, these structural analyses supported that the catalytic mechanism of Dat should use a catalytic triad rather than use a water molecule (Fig. 7).

This study revealed that Ac-CoA plays key roles in progressing the catalytic cycle: (i) binding of the CoA moiety of Ac-CoA to Dat (in the cofactor-binding stage) triggers the structural switch, the formation of the Asp46–Arg153 salt bridge, to induce a conformational change in Dat and create the substrate-binding pocket. (ii) Binding of the acetyl group of Ac-CoA to Dat (in the cofactor-binding stage) induces a loop-to-helix transition of α9. The active site formed by Glu47 on α1 and Ser182 and Ser186 on α7 was fine-tuned through the interactions between α9 and α7. (iii) A bottleneck in the tunnel-shaped cavity in Dat was formed after Ac-CoA binding (in the cofactor-binding stage) but widened after substrate binding (in the substrate-binding stage). (iv) Ac-CoA can eject the products (CoA and acetyl substrate) to restart the next catalytic reaction. This study provides structural details on how Ac-CoA turns on the structural switch to rearrange the conformation and control the progress of the catalytic cycle. Our findings on the catalytic cycle in Dat coordinating the structural switch could serve as an alternative model of research on Ac-CoA-dependent GNAT enzymes.

## Methods

**Expression and protein purification**. A truncated Dat in this study, Dat$_{21-230}$, was constructed to remove disordered N-terminus (residues 1–20) and C-terminus (residues 231–240) to obtain crystals more easily than using full-length Dat[35]. Our previous study revealed that truncated Dat exhibited a similar function and secondary structure to full-length Dat[35]. The Dat gene was inserted into the pGEX-6p3 vector, which contained a GST-tag and a PreScission protease cutting site. The protein was expressed in *Escherichia coli* BL21 (DE3) at 18 °C for 20–24 h in the presence of 1 mM isopropyl β-D-1-thiogalactopyranoside (IPTG). Cells were then harvested through centrifugation at 6000 rpm at 4 °C for 30 min. In addition, cells were resuspended in buffer A (400 mM NaCl, 2.7 mM KCl, 10 mM Na$_2$HPO$_4$, 1.8 mM KH$_2$PO$_4$, 10 mM DTT, 10 mM EDTA, pH 7.0) and lysed using a high-pressure homogenizer (GW Technologies). The supernatant was filtered using a 0.22-μm filter, applied to a 5-mL GSTrap HP column (GE Healthcare) and eluted with buffer B (250 mM NaCl, 2.7 mM KCl, 10 mM Na$_2$HPO$_4$, 1.8 mM KH$_2$PO$_4$, 10 mM DTT, 10 mM EDTA, 12 mM reduced glutathione, pH 7.0). The fractions containing target proteins were collected and dialyzed against 4 L of buffer C (50 mM Tris-HCl, 150 mM NaCl, 1 mM DTT, 1 mM EDTA, and 0.5% glycerol, pH 7.0) at 4 °C for 4 h. The GST-tag was removed by adding PreScission protease (GE Healthcare) at 4 °C for 16 h; then, the buffer was changed to buffer A through dialysis for the second purification. The protein sample was loaded onto the GSTrap HP column again to remove the GST-tag and uncleaved GST-Dat. The flow-through fraction was changed to buffer D (50 mM Tris-HCl, 1 mM DTT, 1 mM EDTA, pH 7.0) through dialysis and was applied to a 5-mL HiTrap Q HP column (GE Healthcare). The target protein was eluted with a linear gradient from 0 M to 0.4 M NaCl in buffer D. The purified Dat was then concentrated to 5–10 mg/mL by Amicon® Ultra 15-mL Centrifugal Filters (10,000 NMWL) (Merck Millipore) and stocked in buffer E (50 mM sodium phosphate, 5 mM DTT, 1 mM EDTA, pH 7.0) for crystallization.

$^{15}$N-labeled Dat was obtained by using the same expression and purification process, except that the medium for cells was M9 medium containing $^{15}$N-NH$_4$Cl as the source of nitrogen. The purified Dat was then concentrated to 5 mg/mL and stocked in NMR buffer (50 mM Tris-HCl, 100 mM NaCl, 1 mM DTT, and 1 mM EDTA, pH 7.0) for NMR titration experiments.

**Crystallization and data collection**. The purified Dat was mixed with different ligands before crystallization, and the final concentration of protein was 0.4 mM (10 mg/ml). For the Dat/CoA/Ac-TRYP ternary complex, Dat was mixed with CoA and Ac-TRYP at a 1:4:4 molar ratio. The crystal of Dat/Ac-CoA/tryptophol was obtained by cocrystallization with Ac-CoA and tryptophol. The Ac-CoA competition experiment was validated by mixing the Dat/CoA/Ac-PEA ternary complex with 20 mM Ac-CoA solution for 30 min.

All crystals were made using the hanging-drop vapor diffusion method with a reservoir (0.1 M imidazole, 1.0 M NaH$_2$PO$_4$, 1.6 M K$_2$HPO$_4$, 0.2 M NaCl, pH 7.0). The crystals were flash-frozen in liquid nitrogen directly, and diffraction data were collected at beamlines BL13B1, BL13C1, and TPS05A1 of NSRRC (Taiwan, ROC.) using an ADSC Quantum-315r CCD or RAYONIX MX-300 HS CCD detector at the National Synchrotron Radiation Research Center (NSRRC) in Taiwan.

**Structure determination**. The diffraction data were processed with HKL2000 (HKL Research Inc. USA)[39]. All the structures were determined using molecular replacement with Phaser-MR (simple interface) of PHENIX (version 1.8.1-1168)[40] using a previously reported Dat/Ac-CoA structure (PDB code: 3TE4) as the search model. After initial refinement, ligand fitting was performed by LigandFit Wizard. Iterative cycles of crystallographic refinement were performed using WinCoot[41] and PHENIX. Molecular graphics of structures and electron density maps were visualized using PyMol Molecular Graphics System (Ver. 2.3 Schrödinger, LLC). The 2mFobs-DFmodel and mFobs-DFmodel maps were displayed at 1σ and 2.5σ levels, respectively. Ramachandran plot statistics of the final models were calculated by PROCHECK[42]. The data collection and refinement statistics are presented in Table 1.

**Isothermal titration calorimetry (ITC) experiment**. All ITC experiments were performed using a MicroCal iTC$_{200}$ System (Malvern Panalytical, UK) at 25 °C. Dat was dialyzed against buffer E (50 mM sodium phosphate, 5 mM DTT, 1 mM EDTA, pH 7.0), and ligands were individually dissolved in buffer E. In cofactor-binding experiments, Ac-CoA in 2.25 mM and CoA in 4.5 mM were titrated individually into 150 μM of Dat. In the competition experiments, Ac-CoA and PEA in 3.5 mM were titrated individually into Dat/CoA/Ac-PEA (ratio 1:7:7) ternary complexes with protein concentrations of 120 μM. Forty microliters of ligand solution was loaded into the stirring syringe and then autoinjected into the sample cell, which contained 280 μL of Dat. Titration experiments consisted of an initial injection (0.2 μL) and 19 consecutive injections, each 2 μL in volume and 4 s in duration, with intervals of 3 min between injections. The results were fitted by a

single-binding-site model using the MicroCal ITC-ORIGIN analysis software. All values correspond to the mean of experiments repeated at least three times.

**NMR titration experiments**. NMR titration experiments using 2D $^1$H-$^{15}$N HSQC spectra were collected at 25 °C on an Avance III HD 850-MHz NMR spectrometer (Bruker, Germany) equipped with a $^1$H/$^{13}$C/$^{15}$N cryoprobe. $^{15}$N-labeled protein was dissolved in NMR buffer (50 mM Tris-HCl, 100 mM NaCl, 1 mM DTT, and 1 mM EDTA, in 90% $H_2O$/10% $D_2O$) containing 1 mM DSS (2,2-dimethyl-2-silapentanesulfonic acid). All ligands were prepared in NMR buffer, and the pH was adjusted to 7 before titration into the protein solution. A stock solution of ligand (50 mM for Ac-CoA, 65 mM for CoA, and 100 mM for Ac-PEA) was added to 200 μM Dat, and a series of $^1$H-$^{15}$N HSQC spectra were collected until no changes in chemical shifts were observed. The final protein/ligand molar ratios for Dat were 1:4 for Ac-CoA and 1:20:20 for CoA/Ac-PEA. To test the competition of Ac-CoA against CoA/Ac-PEA, saturated Dat/CoA/Ac-PEA was titrated with Ac-CoA to a final concentration of 8 mM. Spectra were processed with Topspin 3.5 (Bruker, Germany) and analyzed with NMRFAM-SPARKY[43].

**Circular dichroism (CD) spectroscopy**. CD measurements were performed on 1 μM protein in CD buffer (5 mM sodium phosphate, pH 7.0) with a circular dichroism spectrometer MOS-500 (Bio-Logic, France) over the wavelength range from 195 nm to 260 nm at 25 °C. The pathlength of the cuvette was 1 cm.

**Differential scanning calorimetry (DSC)**. Thermal denaturation curves were acquired on a MicroCal VP-Capillary DSC instrument (Malvern Panalytical, UK). The protein concentrations in each of the four samples were 1 mg/mL in 50 mM sodium phosphate buffer (pH 7.0). For the Dat/Ac-CoA complex, Dat/CoA complex, and Dat/CoA/Ac-PEA complex, the concentrations of ligands were 0.6 mM Ac-CoA, 1.2 mM CoA, and 1.2 mM for both CoA and Ac-PEA, respectively. The experiment was performed in the 10 °C–110 °C temperature range with a scan rate of 200 °C/h. The data were analyzed using MicroCal-enabled Origin software (Malvern Panalytical, UK).

**Statistics and reproducibility**. The protein binding constant ($K_d$), thermal stability, secondary structure, and NMR titration data were measured independently at least three times.

**Reporting summary**. Further information on research design is available in the Nature Research Reporting Summary linked to this article.

## Data availability

All relevant data are available upon reasonable request. Coordinates and structure factors have been deposited in the protein data bank, under accession codes PDB ID 6K80 (Dat/Ac-CoA/Tryptophol complex), 5GI9 (Dat/CoA/Ac-TRYP complex), and 5GI5 (Dat/CoA complex). Source data for Fig. 6a–c are provided in Supplementary Data 1–4.

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

## Acknowledgements
We thank Dr. Kuo-Chang Cheng, Ms Jhen-Ni Liao, and Ms Hsin-Ju Lin of the Institute of Bioinformatics and Structural Biology, National Tsing Hua University, for crystal structure determination of Dat in apo form and binary form. We also thank Dr. Shih-Che Sue of the Institute of Bioinformatics and Structural Biology, National Tsing Hua University, for the discussions of NMR experiments and spectra analysis. We also express our gratitude for the technical services of protein crystallography beamlines provided by the Synchrotron Radiation Protein Crystallography Facility of the National Core Facility Program for Biotechnology, Ministry of Science and Technology, and the National Synchrotron Radiation Research Center. We acknowledge Dr. Shang-Te Danny Hsu, Institute of Biological Chemistry, Academia Sinica, for supporting the Auto PEAQ-DSC data collected at the Biophysics Core Facility, funded by Academia Sinica Core Facility and Innovative Instrument Project (AS-CFII108-111). This study was supported by the Ministry of Science and Technology, Taiwan (MOST grant numbers 106-2311-B-007-004-MY3 to P.-C.L. and 108-2311-B-007-002-MY3 to H.-C.C.).

## Author contributions
I.-C.H. and P.-C.L. conceived and managed the project; C.-Y.W. and Y.-C.Y. performed crystallization and solved the crystal structures with indispensable advice from H.-C.C.; Y.-C.L. performed the cavity analysis; Y.-C.Y. and W.-C.D. performed ITC experiments; C.-Y.W. and W.-C.D. performed NMR experiments with valuable collaboration from Y.-Z.L.; C.-Y.W., I.-C.H., and C.-H.L. analyzed protein structures and NMR spectra of different complexes of Dat; C.-Y.W. and I.-C.H. under the supervision of P.-C.L. interpreted the data, compiled the figures for the paper, and wrote the paper with input from all authors.

## Competing interests
The authors declare no competing interests.
