## [Peer Review File · Communications Biology]

Reviewers' comments:

Reviewer #1 (Remarks to the Author):

Review for: The critical role of acetyl coenzyme A in the catalytic cycle of insect arylalkylamine N-acetyltransferase

Chu-Ya Wu¹, I-Chen Hu¹, Yi-Chen Yang¹, Wei-Cheng Ding¹, Chih-Hsuan Lai¹, Yi-Zong Lee^{1,2}, Yi-Chung Liu³, Hui-Chun Cheng¹, Ping-Chiang Lyu^{1,4*}

This manuscript is potentially interesting but needs some revisions to achieve the quality needed for publication.

Findings of the ordered sequential catalytic mechanism is not really novel or surprising for CoA-dependent enzymes (in my opinion). There for a more detailed look into the structures as well as the mechanism would be more interesting. This in my opinion clearly lacks.

Xr-ay study it self is very nicely done and deserves publication.

Introduction:

- what is the naturally catalyzed reaction of Dat, i.e. what is the natural substrate?
- Reaction scheme would be of great help
- Line 47-74: mainly focussed on structural features instead of proper introduction
- Line 65-84: pdb codes of known related structures would be helpful
- Line 90: is tryptamine the natural substrate? Differences to tryptophol? Reaction scheme with ligands in detail would be advantageous

Results:

- Line 101- 121: not easy to follow since very mixed information
- Line 120: „...location of sulfur atom did not change“ -> in Fig 1b color code for Met121 is very confusing/unusual
- Line 130 and 133: rmsd of 0.88Å is not „significantly different“
- Line 172ff: hard to follow, since I think „CoA moiety“ should better be „adenosine moiety“
- Line 202: for me it is not clear why PEA is used (instead of the natural substrate)
- Line 203-205: Table 2 should be directly placed below Fig 4
- Line 213: again: are TRYP and PEA natural substrates?
- Line 215: no crystal structure of Ac-PEA? Figure 5 a is therefore made of data which are not shown (see also lines 237-241)
- Line 222: „heat change by ITC“: Kd is 107µM, error is 48 µM!?
- Line 236 „in crystallo“? correct expression?
- Line 237-241: experiments cannot be validated since only figures are shown

Discussion:

- Line 260-265: is not a „refined“ propose of the catalytic mechanism in my opinion
- Line 281: a detailed description would need the chemical reaction mechanism and not only „movements“ or „interactions“ without further details
- Line 297: „data not shown“ is a no-go in higher impact journals

Methods:

- Line 329: truncated variant could be explained for better understanding
- Line 334 and 337: concentrations in buffer A and B are strange (403 mM ??; 253 mM??)
- Line 350: concentrations (protein etc.)?
- Line 373: ligand concentrations in a range of 2.25-4.50 mM are vague
- Line 278: biological triplicates?
- Line 385: „a stock solution of ligand (50-100 mM) was added“ is also vague
- Line 392ff: DSC measurements were done only once?

References:

- Out of 32 references only three are recent(2016, 2017, 2018, the others are rather old and the authors should clearly look and cite recent literature on CoA binding enzymes and structures.
- Fig 1a: overlay plus wireframe make it crowded/unclear
- Fig 1b: color code in Met121 confusing
- Fig 2 (all panels): too tiny, too crowded, labels too tiny (more „white space“ than real images); wireframe in panel f ridiculous
- Fig 3: more „white space“ than real images, labels too small; pdb codes of all structures used for that figure should be mentioned in the legend
- Fig 5 a,b,c: as already stated: only images for that, no crystal structure
- Fig 6: a bit too simple...
- Table 2 belongs to Fig 4

Minor:

- Some typos and/or grammar issues

Reviewer #2 (Remarks to the Author):

The manuscript entitled "The critical role of acetyl coenzyme A in the catalytic cycle of insect arylalkylamine N-acetyltransferase" by Wu et al. reports the first crystal structures of Dat/Ac-CoA/tryptophol and Dat/CoA/acetyl-tryptamine ternary complexes, one with a substrate analog and the other with its product, respectively. In addition, the authors also determined the structure of Dat/CoA complex to decipher the role of the acetyl group in the reaction cycle. Together with ITC and NMR studies, the authors were able to study the catalytic cycle of Dat in detail. I find this article to be potentially interesting not only for scientists in the area of melatonin biosynthesis but also for those studying structural enzymology in general.

I find the description of reported structures by the authors good. Results from ITC, NMR and DSC were explained and discussed clearly. As the results are all in support of the reaction model the authors propose, I have no objection to the acceptance of this article for publication in "Communications Biology".

Reviewer #3 (Remarks to the Author):

The manuscript describes a comprehensive work on an enzymatic reaction.

Criticism/comments on the manuscript:

Title: "critical role..." maybe "essential or key role ..."

line 29/30 "... in Drosophila ..." The enzyme is specific for Drosophila? maybe other insects?

line 35 "crystallographic snapshots" - Nowadays with FEL x-ray diffraction timeresolved studies are becoming very important, thus, "snapshot" is a little misleading. It is better to refer to intermediate steps / local minima on the reaction pathway.

In lines 116/117, sulfur methionine is mentioned to interact with aromatic systems - in lines 119/120 Met121 orientation is "altered ... but location of sulfur atom did not change". I understand that the methyl group is rotated by 90° - a minor change. Thus, I assume that Met121 has no key function? In line 168 Met121 seemed to be critical ... seems to be a useless discussion.

line 140, "... extra helix alpha9 was formed ..." how many turns or residues? because this is an important conformational change.

line 121, "interhelix hydrogen bonds" by side chains or? which?

line 146, "salt bridge with the CoA moiety" which?

line 147, carboxylate group

line 157, rms deviation of binding residues ... on all atoms?

lines 179/181, the backbone has been changed? or are there conformational changes of sidechains?

line 185, hydrogens bonds are specific for the donor acceptor atoms, thus, which CoA moiety atoms?

lines 225/226, "entropy factor is related ... repulsive force of the solvent or ligand" all this is not really wrong but at least not exactly correct. I think the authors refer to the hydrophobic effect - textbook knowledge.

line 251, again hydrogen bonds to the CoA moiety - which atoms?

line 251, "resonance of Leu146 disappeared" ? it is elsewhere?

line 296, "... lost its secondary structure ..." tertiary structure? completely unfolded? or which part of the protein?

line 323, "And we hope ..."

line 382, "15N-labeled protein ..." not mentioned in the expression methods.

line 401, entry codes

Overall, the work behind the manuscript is interesting, but it is not inviting to read the structural description of substrate/ligand binding and induced conformational changes.

I suggest a revision ...

--- end of comments ----

June 09, 2020

Reviewers

Communications Biology

Dear Reviewers,

Thank you very much for your in-depth review concerning our manuscript entitled “An essential role of acetyl coenzyme A in the catalytic cycle of insect arylalkylamine N-acetyltransferase” (Manuscript NO.: COMMSBIO-20-0307-T). Those comments are valuable and very helpful for revising and improving our paper. Please find below our point-by-point response to the reviewer’s comments. We have indicated the corresponding revised paragraphs in the manuscript by gray highlight. Appendix files used in the response to reviewers can be found at the following link: <https://figshare.com/s/08f58b56a4ad2780505d>

Thank you again for your advice. We hope that this revised version of the manuscript is now acceptable for publication in *Communications Biology*.

Yours sincerely,

Ping-Chiang Lyu, PhD
National Tsing Hua University
Hsinchu 30013, Taiwan

Response to Comments from the Reviewers

Reviewer #1

“This manuscript is potentially interesting but needs some revisions to achieve the quality needed for publication. Findings of the ordered sequential catalytic mechanism is not really novel or surprising for CoA-dependent enzymes (in my opinion). Therefore, a more detailed look into the structures as well as the mechanism would be more interesting. This in my opinion clearly lacks.”

“X-ray study itself is very nicely done and deserves publication.”

Our response: We are grateful to Reviewer #1’s kind suggestion. Discussion about the relationship of structures and chemical reaction mechanism was added. (see page 13, Lines #321–329).

“Introduction:

1. - *what is the naturally catalyzed reaction of Dat, i.e. what is the natural substrate?”*

“- Reaction scheme would be of great help”

Our response: Thank you for your suggestion.. The naturally catalyzed reaction of Dat was considered to transform dopamine into acetylated dopamine, shown by Karlson & Ammon (1963)¹, and Sekeris & Herrlich (1966)², a key component in the insect cuticle sclerotization process. Dat belong to arylalkylamine *N*-acetyltransferase (AANAT), and the proposed reaction was: acetyl-CoA + 2-arylethylamine → *N*-acetyl-2-arylethylamine + CoA-SH. Reaction scheme undoubtedly is of great help. It helps us to illustrate more clearly on the catalyzed reaction and the chemical structures of these substrates. We have added Scheme 1 in the revised manuscript (page 21).

The natural substrates of Dat are dopamine and tyramine, which were first proposed by Dewhurst et al. (1972)³ using *Drosophila* nervous tissues and whole fly extracts. Researchers also found that isolated Dat from *Drosophila melanogaster* can also catalyze tryptamine, dopamine and serotonin (Hintermann et al. (1995))⁴, and recombinant Dat can catalyze diverse monoamines including tyramine, dopamine, octopamine, tryptamine, and phenylethylamine with high affinity by Dempsey et al. (2014)⁵.

Scheme 1. N-acetylation of monoamines by AANATs

2. “- Line 47-74: mainly focused on structural features instead of proper introduction”

Our response: Thank you for your suggestion. We have added general introduction to physiological functions of insect AANATs, including the substrate types, the effect on insect sclerotization, the potential roles in circadian rhythms and in fatty acid amide biosynthesis. A relevant statement for physiological functions in insect AANATs is given in page 3-4 (Lines #63-74) of the revised manuscript.

3. “- Line 65-84: pdb codes of known related structures would be helpful”

Our response: Thank you for the reminder. We have added PDB codes of known related

structures in the introduction section (see page 4, Lines #77-79).

4. “- Line 90: is tryptamine the natural substrate? Differences to tryptophol? Reaction scheme with ligands in detail would be advantageous”

Our response: Thank you for your question. No report examines whether or not tryptamine is the natural substrate of Dat, but both isolated Dat and recombinant Dat have similar affinity to tryptamine and natural substrate dopamine^{4,5}. In addition, tryptamine is commonly used in AANATs studies from insect to mammal^{4,6-10}. The amino group in tryptamine is replaced as hydroxyl group to form tryptophol (both chemical structures are showed below), therefore tryptophol is a dead end analog of tryptamine for N-acetyltransferase. In this study we used tryptophol to obtain the ternary complex of Dat to mimic the stage before catalysis.

Tryptamine

Tryptophol

“Results:

5. - Line 101- 121: not easy to follow since very mixed information”

Our response: Thank you for your comment. To better understand the comparison of ternary complexes of Dat before and after catalysis, we have added subtitle at the beginning of each paragraph and revised the sentences on page 5-6, Lines #114-146 in the revised manuscript.

6. “- Line 130 and 133: rmsd of 0.88Å is not „significantly different“ “

Our response: Thank you for your comment. Indeed, rmsd below 1 Å of the whole

structure was considered no significant change. We have changed the wording on page 6 (Lines #151-152) in the revised manuscript and compared the RMSD value only on the regions of local conformational changes, where all-atom-based RMSD was 2.24 Å between initial stage and cofactor binding stage, and 0.69 Å between cofactor binding stage and substrate binding stage (see page 7, Lines #157-160 in the revised manuscript).

7. “- Line 172ff: hard to follow, since I think „CoA moiety“ should better be „adenosine moiety“”

Our response: Thank you for your comment. The reason why we used “CoA moiety” was to distinguish acetyl group and CoA part of Ac-CoA molecule on influencing Dat’s conformation by comparing two crystal structures, Dat/Ac-CoA and Dat/CoA. We have also simplified our paragraph for clear understanding (see page 8-9, Lines #205-216 and Fig. 3 in the revised manuscript).

8. “- Line 202: for me it is not clear why PEA is used (instead of the natural substrate)”

“- Line 203-205: Table 2 should be directly placed below Fig 4”

Our response: Thank you for your questions. The reason why we used PEA as a substrate instead of using others is that: (i) PEA is easy to be obtained and stable. (ii) Most substrates for Dat are based upon the PEA structure as the core structure with substituents replacing one or more hydrogens (see page 21, Scheme 1 in the revised manuscript). We have placed the table directly below Fig. 4 on page 26 in the revised manuscript. Thank you for your advice.

10. “- Line 213: again: are TRYP and PEA natural substrates?”

Our response: Thank you for your question. No report directly mentions that TRYP and PEA are natural substrates of Dat. The reason for why we choose TRYP and PEA in this study please see our previous response.

11. “- Line 215: no crystal structure of Ac-PEA? Figure 5 a is therefore made of data which are not shown (see also lines 237-241)”

Our response: Thank you for your kind reminder. We have added the PDB code (5GI7) of the crystal structure of Ac-PEA (see page 9, Line #222 in the revised manuscript) In addition, the crystal raw data corresponded in original Fig. 5a-c were attached subsequently in Appendix 1:

1. Data collection and refinement statistics
2. Structures (pdb files) and structure factors (mtz files)
3. Calculated mFo-DFc_map coordinating with the structures (pymol file)

12. “- Line 222: „heat change by ITC“: K_d is $107\mu M$, error is $48\mu M$!?”

Our response: Thank you for your correction. For reducing the error values of K_d in ITC experiments, we have repeated more times of the titrations of Ac-CoA to Dat/CoA/Ac-PEA (n=6) and re-calculated the standard deviation of K_d (see Fig. 4 on page 26 in the revised manuscript). The raw data of all the titrations in Fig. 4a-4d were shown in Appendix 2.

13. “- Line 236 „in crystallo“? correct expression?”

Our response: Yes, we found this word in the literature to express the observation of ligand and protein using crystallographic methods. In our study, we monitored the changes of ligands and protein in crystallo (by crystal structure) and in solution (by NMR).

There are 131 literature contains “in crystallo” in the search result of PubMed. The following are three examples:

1. In Crystallo Thermodynamic Analysis of Conformational Change of the Topaquinone Cofactor in Bacterial Copper Amine Oxidase. (2019) PNAS, 116(1):135-140
2. In Crystallo Selection to Establish New RNA Crystal Contacts. (2018), Structure, 26(9):1275-1283
3. In Crystallo Capture of a Covalent Intermediate in the UDP-Galactopyranose Mutase

Reaction. (2016) *Biochemistry*, 55(6):833-6
Observing cellulose biosynthesis and membrane translocation in crystallo. (2016) *Nature*, 531:329–334

14. “- *Line 237-241: experiments cannot be validated since only figures are shown*”

Our response: Thank you for your suggestion. The crystal raw data corresponded in original Fig. 5a-c were attached as follows in Appendix 1:

1. Data collection and refinement statistics
2. Structures (pdb files) and structure factors (mtz files)
3. Calculated mFo-DFc_map coordinating with the structures (pymol file)

“Discussion:

15. - *Line 260-265: is not a „refined“ propose of the catalytic mechanism in my opinion*”

“- Line 281: a detailed description would need the chemical reaction mechanism and not only „movements“ or „interactions“ without further details”

Our response: We thank Reviewer #1 for the valuable suggestions. We have changed the words “catalytic mechanism” into “catalytic cycle” to avoid misleading (see page 10, Line #259 in the revised manuscript). About the detailed description of chemical reaction mechanism, we had introduced it in page 4, Line #85-93. In addition, we have added one paragraph to discuss the catalytic mechanism of Dat (see page 13, Lines #321-329), and added the illustration of the chemical reaction mechanism in Fig. 6 (see page 28 in the revised manuscript).

16. “- *Line 297: „data not shown“ is a no-go in higher impact journals”*”

Our response: Thank you for your advice. We have supplemented the data about CD spectra and NMR HSQC spectra of the mutant M121A in Supplementary Fig. 3 in the revised Supplementary information, to show a single mutation influence the stability of the protein. (Also see page 12, Lines #298-302 in the revised manuscript).

” Methods:

17. “- *Line 329: truncated variant could be explained for better understanding*”

Our response: Thank you for your suggestion. The truncated variant used in this study, Dat₂₁₋₂₃₀, does not have the disordered N-terminus (residues 1-20) and C-terminus (residues 231-240) and helps us more easily obtain the crystals than using Dat of full length. Our laboratory constructed and studied Dat₂₁₋₂₃₀ for several years, and confirmed that the enzymatic characteristics of this truncated variant were similar to those of full-length Dat. We have revised the sentence (see page 13, Lines #344-345 in the revised manuscript.)

18. “- *Line 334: and 337: concentrations in buffer A and B are strange (403 mM ??; 253 mM??)*

“- *Line 350: concentrations (protein etc.)?*”

“- *Line 373: ligand concentrations in a range of 2.25-4.50 mM are vague*”

“- *Line 385: „a stock solution of ligand (50-100 mM) was added“ is also vague*”

Our response: Thank you for your suggestion. We used 10x PBS buffer and 5 M sodium chloride solution to prepared buffer A and buffer B. The presented strange concentrations were caused by the misunderstanding of the concentration of sodium chloride in PBS. It is exact 1.37 M instead of 1.40 M in 10x PBS stock. **We have revised the concentrations of NaCl to 400 mM and 250 mM for buffer A and buffer B, respectively** (see page 14, Lines #350 and 353 in the revised manuscript). We apologize for this kind of mistake. For other comments (Line 350, 373, and 385 in the original version) about experimental conditions, we have revised the concentrations of protein and ligands in detail (see page 14-16 Lines #371-372, Lines #394-398, and Lines #409-410 in the revised manuscript)

19. “- *Line 278: biological triplicates?*”

Our response: Thank you for your suggestion. For the comment about ITC measurements, all values correspond to the mean of experiments which repeated at least 3 times with different lot of proteins and the raw data of all the titrations in ITC were

shown in Appendix 2 (see page 15, Lines #401-402 and Appendix 2).

20. “- *Line 392ff: DSC measurements were done only once?*”

Our response: Thank you for your suggestion. For DSC measurements, all measurements were done by three times with the calculated standard deviation of T_m value in Supplementary Fig. 4 in the revised Supplementary information.

“References:

21. - *Out of 32 references only three are recent (2016, 2017, 2018, the others are rather old and the authors should clearly look and cite recent literature on CoA binding enzymes and structures.*”

“- *Fig 1a: overlay plus wireframe make it crowded/unclear*”

“- *Fig 1b: color code in Met121 confusing*”

“- *Fig 2 (all panels): too tiny, too crowded, labels too tiny (more „white space“ than real images); wireframe in panel f ridiculous*”

“- *Fig 3: more „white space“ than real images, labels too small; pdb codes of all structures used for that figure should be mentioned in the legend*”

“- *Fig 5 a,b,c: as already stated: only images for that, no crystal structure*”

“- *Fig 6: a bit too simple...*”

“- *Table 2 belongs to Fig 4*”

Our response: We agree with Reviewer #1’s comment that we should clearly look more literature on CoA binding enzymes and structures. We have revised our manuscript and supplemented the literature review in the introduction section. Also, to make figures more clearly and easily to read, we have checked and redrawn all figures. We have replaced the wireframe by only gray color in Fig. 1a, and changed the color code of residues in Fig. 1b. We have simplified and summarized Fig. 2, Fig. 3 and Fig. 4, therefore the corresponding paragraphs have been slightly modified (please see page 6-

11, Lines #148-203, #218-232, #265-270, and #283-293 in the revised manuscript). We have rearranged the typesetting of Fig. 4 with the table. Regarding the crystal data in Fig. 5a-c, we have attached raw data in Appendix 1. Furthermore, in Fig. 6, as your suggestion, we have added chemical reaction mechanism to the catalytic cycle of Dat. Thank you for your suggestions.

“Minor:

22. - Some typos and/or grammar issues”

Our response: Thanking you for pointing this mistake, we have corrected typos and grammar issues in the revised manuscript. We also have sent our manuscript to English editing to proofread and strengthen our English text.

Reviewer #2

“The manuscript entitled "The critical role of acetyl coenzyme A in the catalytic cycle of insect arylalkylamine N-acetyltransferase" by Wu et al. reports the first crystal structures of Dat/Ac-CoA/tryptophol and Dat/CoA/acetyl-tryptamine ternary complexes, one with a substrate analog and the other with its product, respectively. In addition, the authors also determined the structure of Dat/CoA complex to decipher the role of the acetyl group in the reaction cycle. Together with ITC and NMR studies, the authors were able to study the catalytic cycle of Dat in detail.”

“I find this article to be potentially interesting not only for scientists in the area of melatonin biosynthesis but also for those studying structural enzymology in general.”

“I find the description of reported structures by the authors good.”

“Results from ITC, NMR and DSC were explained and discussed clearly. As the results are all in support of the reaction model the authors propose, I have no objection to the acceptance of

this article for publication in "Communications Biology".

Our response: We thank Reviewer #2 for appreciating our work and supporting it for publication in *Communications Biology*.

Reviewer #3

"The manuscript describes a comprehensive work on an enzymatic reaction."

"Overall, the work behind the manuscript is interesting, but it is not inviting to read the structural description of substrate/ligand binding and induced conformational changes."

"Criticism/comments on the manuscript:

1. Title: "critical role..." maybe "essential or key role ..."

Our response: We are grateful to Reviewer #3 for the comments and have modified the title accordingly as suggested (see page 1, Line #2 in the revised manuscript).

2. "line 29/30 "... in *Drosophila* ..." The enzyme is specific for *Drosophila*? maybe other insects?"

Our response: Thank you for your suggestion. Dopamine N-acetyltransferase (Dat) is not specific for *Drosophila*. By searching the protein sequence on UniProt website can find the protein exist in a lot of insects, such as *Lucilia cuprina*, *Culex quinquefasciatus*, *Anopheles darling*, *Asbolus verrucosus*, and *Apis cerana cerana*, with sequence identities of 61.8%, 58.7%, 57.7%, 44.2%, and 38.7%, respectively.

Besides, there are eight putative AANAT-like proteins found in *Drosophila melanogaster*^{5,11}, among them Dat is the first identified insect AANAT from *Drosophila melanogaster*^{4,12,13}.

3. "line 35 "crystallographic snapshots" - Nowadays with FEL x-ray diffraction time resolved

studies are becoming very important, thus, "snapshot" is a little misleading. It is better to refer to intermediate steps / local minima on the reaction pathway.””

Our response: Thank you for your suggestion. Indeed, time resolved-FEL x-ray diffraction provides a serial snapshot for the dynamical changes and catalysis in enzyme. We thank you for the correction and have changed the wording as suggested (see page 2, Lines #34-35 in the revised manuscript).

4. *“In lines 116/117, sulfur methionine is mentioned to interact with aromatic systems”*

“- in lines 119/120 Met121 orientation is "altered ... but location of sulfur atom did not change". I understand that the methyl group is rotated by 90° - a minor change. Thus, I assume that Met121 has no key function?”

“In line 168 Met121 seemed to be critical ... seems to be a useless discussion.

Our response: Thank you for your suggestion. We reviewed the performance of Met121 in apo form and complex form structures, considering that the behaviors of residue side chain in crystal and in solution may be different, so we deleted the description and discussion about the role of Met121 on the regulation of the bottleneck.

5. *“line 140, "... extra helix alpha9 was formed ..." how many turns or residues? because this is an important conformational change.”*

“line 142, "interhelix hydrogen bonds" by side chains or? which?”

Our response: Thank you for your suggestion. An extra helix $\alpha 9$ was formed through five residues Ala217, Ala218, Pro219, His220, and Val221. The interhelix hydrogen bonds between $\alpha 7$ and $\alpha 9$ were formed by the side chain of Ser183 and the side chain of His220, as well as the backbone amide of His184 and the backbone carbonyl of His220. We have added the detail of hydrogen bond interactions between $\alpha 7$ and $\alpha 9$ in the revised manuscript (see page 7, Lines #156-157 and Lines #175-176).

6. *“line 146, "salt bridge with the CoA moiety" which?”*

“line 147, carboxylate group”

Our response: Thank you for the reminder and correction. The salt bridges were formed through the pyrophosphate group and the 3'-phosphate group of CoA moiety to the side chain of Lys192. We also have corrected the word “carboxylate group” (see page 7, Lines #165-166 and Lines # 169-170 in the revised manuscript)

7. *“line 157, rms deviation of binding residues ... on all atoms?”*

Our response: Thank you for your question. Regarding the RMSD of the binding residues, we calculated on all atoms of the binding residues. However, we delete the sentence to reduce the complicated information in that paragraph in the revised manuscript.

8. *“lines 179/181, the backbone has been changed? or are there conformational changes of sidechains?”*

Our response: Thank you for your comment. The position of $\alpha 7$ in Dat/Ac-CoA complex was significantly different from those in Dat/CoA complex and Apo-Dat. Therefore, both backbones and side chains, the positions of Ser182 (the residues on the loop before $\alpha 7$) and Ser186 (on $\alpha 7$) were different to the corresponding residues in Dat/CoA complex.

9. *“line 185, hydrogens bonds are specific for the donor acceptor atoms, thus, which CoA moiety atoms?”*

Our response: Thank you for your question. Several hydrogen bonds formed among the backbone atoms of a region from Leu146 to Ala158 Ac-CoA or CoA. The details were listed in Supplementary Table 2 (see page 7, Lines #163-165 in the revised manuscript). Here we also wanted to focus on the salt bridge formed by the side-chain nitrogen atom of Lys192 to the oxygen atoms in pyrophosphate groups and the 3'-phosphate group of the CoA group.

10. *“lines 225/226, “entropy factor is related ... repulsive force of the solvent or ligand” all this is not really wrong but at least not exactly correct. I think the authors refer to the hydrophobic effect - textbook knowledge.”*

Our response: Thank you for the correction. We have tried our best to find another reference to describe the entropy factor. In the new reference, the author indicated that the entropy factor also comes from the conformational entropy change¹⁴. The conformational entropy change comes from the degrees of freedom of both protein and ligand. Thus, we thought that the increasing entropy in our experiment of competition reaction of Ac-CoA against the product ternary complex may imply the release of products. However, to avoid ambiguous description, we have deleted that part in the revised manuscript. Thank you for your comment.

11. *“line 251, again hydrogen bonds to the CoA moiety - which atoms?”*

Our response: Thank you for the comment. The hydrogen bond formed between the backbone carbonyl oxygen atom of Leu146 and the the amide hydrogen atom of β -mercaptoethylamine moiety of Ac-CoA and CoA. (see page 10, Lines #248-250 in the revised manuscript).

12. *“line 251, “resonance of Leu146 disappeared”? it is elsewhere?”*

Our response: Thank you for your question. We have checked the resonance of Leu146 by backbone assignments of Dat/CoA/Ac-PEA complex, end of Ac-CoA titration, and Dat/Ac-CoA complex, however, no resonance corresponding to Leu146 was found, suggesting that multiple states of Leu146 existed during the titration of Ac-CoA because Leu146 can interact with Ac-PEA, CoA, and Ac-CoA. The results of backbone assignments were supplemented in Appendix 3.

13. *“line 296, “... lost its secondary structure ... tertiary structure? completely unfolded? or which part of the protein?”*

Our response: Thank you for your questions. Our data showed that mutant M121A lost part of secondary structure (from CD spectra) and tertiary structure (from NMR HSQC spectra) and presented in an unstable structure that easily prompts to aggregate. We have supplemented the CD spectra and NMR HSQC spectra of mutant M121A in

Supplementary Fig 3 in the revised Supplementary information, to show a single mutation influence the structural stability of the protein. (Also see page 12, Lines #298-302 in the revised manuscript).

14. “line 323, “And we hope ...””

Our response: We have rewritten our sentence, and we hope this would be more suitable. Thank you for your comment (see Lines #339-340 in the revised manuscript).

15. “line 382, “¹⁵N-labeled protein ...” not mentioned in the expression methods.”

Our response: Thank you for your reminder. The expression method for ¹⁵N-labeled protein have be written in page 14, Lines #365-368 in the revised manuscript.

16. “line 401, entry codes”

Our response: We apologize for our carelessness and thank the reviewer for pointing this out. We have corrected the word in page 16, Line #430 in the revised manuscript.

REFERENCES

1. Karlson, P. & Ammon, H. Zum Tyrosinstoffwechsel Der Insekten .11. Biogenese Und Schicksal Der Acetylgruppe Des N-Acetyl-Dopamins. *Hoppe-Seylers Zeitschrift Fur Physiologische Chemie* **330**, 161-& (1963).
2. Sekeris, C.E. & Herrlich, P. Zum Tyrosinstoffwechsel Der Insekten .17. Der Tyrosinstoffwechsel Von Tenebrio Molitor Und Drosophila Melanogaster. *Hoppe-Seylers Zeitschrift Fur Physiologische Chemie* **344**, 267-& (1966).
3. Dewhurst, S.A., Ikeda, K., Mccaman, R.E. & Croker, S.G. Metabolism of Biogenic-Amines in Drosophila Nervous-Tissue. *Comparative Biochemistry and Physiology* **43**, 975-& (1972).
4. Hintermann, E., Jenö, P. & Meyer, U.A. Isolation and Characterization of an Arylalkylamine N-Acetyltransferase from Drosophila-Melanogaster. *Febs Letters* **375**, 148-150 (1995).
5. Dempsey, D.R. et al. Mechanistic and Structural Analysis of Drosophila melanogaster Arylalkylamine N-Acetyltransferases. *Biochemistry* **53**, 7777-7793 (2014).
6. De Angelis, J., Gastel, J., Klein, D.C. & Cole, P.A. Kinetic analysis of the catalytic mechanism of serotonin N-acetyltransferase (EC 2.3.1.87). *Journal of Biological*

- Chemistry* **273**, 3045-3050 (1998).
7. Aboalroub, A.A. et al. Acetyl group coordinated progression through the catalytic cycle of an arylalkylamine N-acetyltransferase. *Plos One* **12**(2017).
 8. Hickman, A.B., Namboodiri, M.A.A., Klein, D.C. & Dyda, F. The structural basis of ordered substrate binding by serotonin N-acetyltransferase: Enzyme complex at 1.8 angstrom resolution with a bisubstrate analog. *Cell* **97**, 361-369 (1999).
 9. O'Flynn, B.G. et al. Characterization of Arylalkylamine N-Acyltransferase from *Tribolium castaneum*: An Investigation into a Potential Next-Generation Insecticide Target. *Acs Chemical Biology* **15**, 513-523 (2020).
 10. Tsugehara, T., Imai, T. & Takeda, M. Characterization of arylalkylamine N-acetyltransferase from silkworm (*Antheraea pernyi*) and pesticidal drug design based on the baculovirus-expressed enzyme. *Comparative Biochemistry and Physiology C-Toxicology & Pharmacology* **157**, 93-102 (2013).
 11. Amherd, R., Hintermann, E., Walz, D., Affolter, M. & Meyer, U.A. Purification, cloning, and characterization of a second arylalkylamine N-acetyltransferase from *Drosophila melanogaster*. *DNA and Cell Biology* **19**, 697-705 (2000).
 12. Maranda, B. & Hodgetts, R. Characterization of Dopamine Acetyltransferase in *Drosophila-Melanogaster*. *Insect Biochemistry* **7**, 33-43 (1977).
 13. Brodbeck, D. et al. Molecular and biochemical characterization of the aaNAT1 (Dat) locus in *Drosophila melanogaster*: Differential expression of two gene products. *DNA and Cell Biology* **17**, 621-633 (1998).
 14. Freire, E. Isothermal titration calorimetry: controlling binding forces in lead optimization. *Drug Discov Today Technol* **1**, 295-9 (2004).

REVIEWERS' COMMENTS:

Reviewer #1 (Remarks to the Author):

Dear Authors,

I greatly appreciated the changes you made in the revised version of your manuscript. I especially liked the new figure 5 with the electron density.

I am in favour that this manuscript is published.

Two little comments I have which can be made during the typesetting.

1) please add the number of experiments ($n=6$) in the legend of the ITC figure. This to make sure people understand that the K_d values are performed and calculated from replicates.

2) Figure 1 left panel. In my view I have the feeling that the labels Tryptophol and Ac-Tryp are switched. But maybe that is my mistake. Please have a look and change when I am correct.

All other points are very well addressed and personally I really like the manuscript. Well done.

Reviewer #3 (Remarks to the Author):

Dear authors,

I have looked to the three referee reports, your comments, and traveled through the revised manuscript.

I think the improved version is acceptable.

Minor comment "pyrophosphate" is old fashioned, "diphosphate" is (slightly) better.

--- end of comments ---

July 16, 2020

Reviewers

Communications Biology

Dear Reviewers,

Thank you very much for all the valuable comments. Without your advice, we cannot revise flaws with more rigorous academic standard in our revised manuscript. Please find below our response to the reviewer's comments. We have indicated the corresponding revised sentences in the manuscript by gray highlight.

Thank you again for the advice from you.

Yours sincerely,

Ping-Chiang Lyu, PhD
National Tsing Hua University
Hsinchu 30013, Taiwan

=====

Response to Comments from the Reviewers

Reviewer #1

“Dear Authors,

I greatly appreciated the changes you made in the revised version of you manuscript.

I especially liked the new figure 5 with the electron density.

I am in favour that this mansucript is published.

Two little comment I have which can be made during the typesetting.

1) please add the number of experiment (n=6) in the legend of the ITC figure. This to make sure people understand that the Kd values are performed and calculated from replicates.

2) Figure 1 left panel. In my view I have the feeling that the label Tryptophol and Ac-Tryp are switched. But maybe that is my mistake. Please have a look and change when I am correct.

All other points are very well addressed and personally I really like the manuscript.

Well done.”

Our response: We are very grateful for Reviewer #1’s kind suggestion and appreciation of our revision. We have added the number of experiments in the legend as your suggestion. (see Figure5 and Table 2 in the revision). We have double checked the label in Figure 1. Thank you so much again for all the time and effort you put in reviewing our paper and sharing your valuable experiences.

Reviewer #3

“Dear authors,

I have looked to the three referee reports, your comments, and traveled through the revised manuscript.

I think the improved version is acceptable.

Minor comment "pyrophosphate" is old fashioned, "diphosphate" is (slightly) better.”

Our response: We thank very much for Reviewer #3’s kind suggestion and support for the publication. We have used the word "diphosphate" instead of "pyrophosphate". (see page 7, Line #168 and 174 in the revised manuscript). We really appreciate your time and effort, which have greatly enhanced our work.